# Community-Based Approaches to Increase COVID-19 Vaccine Uptake and Demand: Lessons Learned from Four UNICEF-Supported Interventions

**DOI:** 10.3390/vaccines11071180

**Published:** 2023-06-30

**Authors:** Kathryn L. Hopkins, Talya Underwood, Iddi Iddrisu, Hanna Woldemeskel, Helena Ballester Bon, Symen Brouwers, Sofia De Almeida, Natalie Fol, Alka Malhotra, Shalini Prasad, Sowmyaa Bharadwaj, Aarunima Bhatnagar, Stacey Knobler, Gloria Lihemo

**Affiliations:** 1Sabin Vaccine Institute, Washington, DC 20037, USA; kate.hopkins@sabin.org (K.L.H.);; 2Sabin Vaccine Institute, Northwich CW9 7DA, UK; 3UNICEF Ghana Country Office, Accra-North P.O. Box 5051, Ghana; iiddrisu@unicef.org; 4UNICEF Zambia Country Office, Lusaka P.O. Box 33610, Zambia; 5UNICEF Regional Office for East and Southern Africa, Nairobi P.O. Box 44145, Kenya; 6UNICEF India Country Office, New Delhi 110 003, India; 7Praxis Institute for Participatory Practices, New Delhi 110 016, India; 8UNICEF Iraq Country Office, Baghdad 10011, Iraq; 9UNICEF HQ, New York, NY 10017, USA

**Keywords:** COVID-19, vaccines, pandemic, routine immunization, vaccine confidence, vaccine hesitancy, risk communication, lessons learned

## Abstract

Vaccination is critical to minimize serious illness and death from COVID-19. Yet uptake of COVID-19 vaccines remains highly variable, particularly among marginalized communities. This article shares lessons learned from four UNICEF interventions that supported Governments to generate acceptance and demand for COVID-19 vaccines in Zambia, Iraq, Ghana, and India. In Zambia, community rapid assessment provided invaluable real-time insights around COVID-19 vaccination and allowed the identification of population segments that share beliefs and motivations regarding COVID-19 vaccination. Findings were subsequently used to develop recommendations tailored to the different personas. In Iraq, a new outreach approach (3iS: Intensification of Integrated Immunization) utilized direct community engagement to deliver health messages and encourage service uptake, resulting in over 4.4 million doses of COVID-19 and routine immunization vaccines delivered in just 8 months. In Ghana, a human-centered design initiative was applied to co-develop community-informed strategies to improve COVID-19 vaccination rates. In India, a risk communication and community engagement initiative reached half a million people over six months, translating into a 25% increase in vaccination rates. These shared approaches can be leveraged to improve COVID-19 vaccination coverage and close gaps in routine immunization across diverse and marginalized communities.

## 1. Introduction

COVID-19 vaccination remains a vital tool to protect the health of the global population and reduce serious illness and death from the virus. However, three years into the pandemic, uptake of COVID-19 vaccines remains highly variable, as a result of persistent inequities in vaccine access, and growing misinformation around vaccination. While 70% of the world’s population has received at least one COVID-19 vaccine (12 billion doses have been administered globally), coverage within low- and middle-income countries (LMICs) is just 29% [1]. These settings, comprising marginalized communities, have been disproportionately affected by inequities around COVID-19 vaccination. There are additional challenges in reaching priority groups who remain the most vulnerable (e.g., children, immunocompromized individuals and those who are at high-risk of infection, and historically marginalized communities) and inadequate data to inform targeted interventions to reach them.

The COVID-19 pandemic also severely affected routine immunization programs, with many countries observing declines in vaccination coverage [2]. Between 2019 and 2021, 67 million children have missed out either entirely or partially on routine childhood immunizations, leaving them at risk of preventable illness and death [3]. The pandemic has also driven a rise in misinformation around vaccines, in general, and in some situations, reduced trust in health services [4,5]. Other emergencies, such as civil unrest and natural disasters linked to climate change, have caused further disruptions to health service delivery, creating competing priorities.

In moving forward, minimizing the impact of COVID-19 will require efforts to close critical immunization gaps. This will require a balanced approach that integrates COVID-19 into routine vaccination and the wider primary health care system. Broadly, this calls for multidisciplinary health system strengthening approaches that render vaccination easy and convenient and promote pro-equity, gender-sensitive, evidence-based tailored strategies to reach the most vulnerable.

As the factors affecting acceptance and demand for vaccines are complex and often context-specific, a range of tools and strategies are needed to implement contextually relevant solutions to promote confidence in and uptake of COVID-19 vaccines around the world. In this context, community-driven interventions are key to developing solutions that are evidence-based, context-specific, and culturally appropriate to drive impactful, sustainable change around vaccination. People-centered approaches are also critical to collectively diagnose uptake barriers and co-design solutions with communities.

A key component of UNICEF’s work is leveraging social and behavior change principles that blend scientific knowledge with community insights, to bring about positive change [6]. UNICEF and its partners work to develop evidence-based approaches that support Governments to strengthen immunization programs and drive demand for and uptake of vaccines across diverse settings.

Here we share lessons learned from four case studies of UNICEF-driven country interventions that support Governments to generate acceptance and demand for COVID-19 vaccines and routine immunization in Zambia, Iraq, Ghana, and India. Four key approaches rooted in social and behavioral sciences were deployed: community rapid assessment, 3iS (Intensification of Integrated Immunization) outreach, human-centered design, and risk communication and community engagement (RCCE). The article describes how these approaches were successfully used to improve COVID-19 vaccination coverage, close gaps in routine immunization, and strengthen pandemic preparedness efforts among diverse and marginalized communities.

## 2. Community Interventions to Generate Acceptance and Demand for COVID-19 Vaccines

### 2.1. Zambia Case Study: Utilizing Community Rapid Assessment for Real-Time Insights into COVID-19 Knowledge, Behavior, and Demand for Vaccines

#### 2.1.1. Background

Zambia has experienced four devastating waves of the COVID-19 pandemic, with each wave amplifying in magnitude in terms of infectivity and mortality. The country initiated a nine-pronged strategic response for the COVID-19 pandemic. Within this, RCCE plays a critical role in strengthening community engagement, providing accurate and timely information, and promoting vaccine acceptability through addressing myths and rumors, and driving uptake of COVID-19 vaccines and essential health services.

Since the introduction of COVID-19 vaccines in Zambia in early 2021, uptake has been slow due to overwhelming misinformation, as well as limited access to vaccines. The coordinated rollout of the national Advocacy Communication Social Mobilization strategy for vaccine demand played a major role in improving COVID-19 vaccine uptake. However, misinformation and concerns related to vaccine efficacy and fear of side effects continued to challenge steady uptake and achievement of expected targets. Demand for essential services like immunization, and maternal and newborn health services was also affected.

In this setting, social and behavior change (SBC) interventions are powerful approaches for influencing drivers of change and supporting local action towards better societies [6]. UNICEF employs social and behavioral evidence and inclusive, participatory approaches to work with communities to understand what drives decision-making, and design interventions that create meaningful, sustainable change in the health sector. Central to this approach is recognizing that populations are heterogeneous, and people have different attitudes, motivations and practices around topics such as vaccination. For example, some people may not trust in the efficacy of the vaccines or are worried about long-term side effects, while others may largely trust vaccines but face difficulties in accessing vaccination services. Consequently, SBC interventions are most effective when they are tailored to different segments of society and address the specific needs of different groups. Real-time insights into individual knowledge, perceptions, motivations, and practices around the prevention of COVID-19, as well as the environmental or contextual factors that affect uptake, are therefore important to better understand the drivers of vaccine acceptance and demand. In this context, community rapid assessment provides a way to gather key data from the community and design tailored community-informed solutions.

#### 2.1.2. Approach

UNICEF, in collaboration with Zambia’s Ministry of Health, deployed community rapid assessment to gather key information about vaccine acceptance and demand. The assessment employed the behavioral and social drivers (BeSD) of vaccination framework, established by the World Health Organization [7]. This measures four domains influencing vaccine uptake: what people think and feel about vaccines; social processes that drive or inhibit vaccination; individual motivations (or hesitancy) to seek vaccination; and practical factors involved in seeking and receiving vaccination [7]. The findings were subsequently used to identify population-wide personas that similarly represent the vaccination-related constructs of ‘thinking and feeling’, ‘social norms’, and ‘levels of motivation’.

A mobile survey via interactive voice response (IVR) was developed to establish ongoing evidence on key behaviors around COVID-19 prevention and insights into demand for COVID-19 vaccines. The survey was designed and executed by Viamo, a global social enterprise specializing in mobile engagement and information and communication technology for development. A rapid assessment was implemented over two rounds, between 30 March to 25 April 2022, (Round 1) and from 8–16 October 2022, (Round 2). Eligible participants included any adult in Zambia from Viamo’s database who opted into surveys.

#### 2.1.3. Results

Over 50,000 individuals from Viamo’s database were contacted across the two rounds, with *n* = 1970 respondents in Round 1 and *n* = 1210 in Round 2). There were 72.8% and 57.7% male respondents in Round 1 and Round 2, respectively; with 64.1% from urban areas in the first round compared with 37.0% from the second round. Study participants resided across all ten provinces in Zambia, with the majority from Lusaka, the capital and largest city in the country.

Regarding the COVID-19 vaccine and intentions, in Round 1, 44.3% of participants received the first dose, 27.9% were fully vaccinated, and 6.9% received the booster. Among the unvaccinated, 47.6% had plans to get vaccinated. However, unvaccinated females were slightly more hesitant than unvaccinated men, as well as respondents in urban versus rural areas. In Round 2, 41.5% of participants considered the vaccine to be very important. In terms of access to the vaccine, 56% thought it was very easy to access and over 70% thought close family and relatives wanted them to get vaccinated. Overall, 36.2% were fully vaccinated, and 12.5% received the booster. Among the unvaccinated, 47.5% had plans to get vaccinated. Most respondents (82.1%) had sufficient information on where and how to access COVID-19 vaccination and 72.8% said they understood the side effects. The most trusted sources of information about COVID-19 and the COVID-19 vaccine were health workers/health facilities, followed by electronic media.

Drivers of vaccination by gender and location are shown in Figure 1. The majority of female and male respondents reported ‘very easy’ trust in the vaccine (31% and 34%), belief in the vaccination-related benefits for health (40% and 43%), and ease of access to vaccination (57% and 56%). However, there was a far larger percentage of respondents sharing ‘very easy’ access to vaccination as compared with ‘very easy’ trust in the vaccine (Figure 1a). There were only small differences in findings between female and male respondents, suggesting little impact of gender on drivers of vaccination. A similar trend was seen across rural and urban participants, with few differences between respondents in urban and rural locations (Figure 1b). Together, the findings suggest that decision-making cuts across demographics.

Further analysis of the survey findings identified four persona types in the Zambian population: *Converts*—those who would quickly accept a COVID-19 vaccine; *Outsiders*—those who were unsure about their feelings around COVID-19 vaccination; *Dissenters*—those not convinced about perceived threat of COVD-19 and the need for vaccination and highly susceptible to misinformation; and *Analysts*—individuals who review multiple sources of information and are likely to be highly accepting of vaccination (Table 1). Converters made up the majority of the Zambian population (57%, Round 2), while one-fifth of respondents were categorized as Outsiders. A minority of respondents shared Dissenter personas.

Specific SBC interventions were subsequently designed for the different personas, summarized in Table 2.

#### 2.1.4. Key Lessons Learned

The community rapid assessment provided a wealth of insights into the barriers and drivers around COVID-19 vaccine acceptance and demand. The survey also enabled the identification of population-wide personas that share thinking and feeling, social norms, and levels of motivation. This allowed the development of tailored strategies to improve vaccine acceptance and demand for the different personas and informed the revision of the national Advocacy Communication Social Mobilization strategy. Overall, the adoption of an evidence-driven demand generation approach by the Ministry of Health was a game changer for understanding the barriers for vaccine uptake and designing tailored SBC interventions.

Despite the population share of Converts rising 16% between survey rounds, trust in vaccine safety and efficacy declined. This highlights the importance of sharing timely fact-based information about such topics through trusted sources, which should be maximized using interactive/dialogue-based multimedia and partnering with health professional associations to positively shift vaccination perceptions within the health worker community and the public. It is important to increase tailored messages and interventions to address safety concerns and the information gap existing amongst women and key vulnerable groups and within urban locations. Efforts should be made to increase access to vaccination for those who are willing to get the vaccine but find it hard for various reasons (e.g., providing simplified registration process and implementing education and outreach, including information on vaccination accessibility).

In moving forward, investing in qualitative SBC data collection to understand reasons behind this decline in trust should be prioritized. Priority attention should also be given to reinforcing trust building interventions, such as engaging religious leaders and key influencers as champions, and engaging social networks of vulnerable groups. Greater representation of women should be ensured in subsequent surveys to understand specific gender-related barriers. Moreover, as COVID-19 vaccine provision is increasingly integrated into routine services, a targeted approach will remain critical to increase uptake among such vulnerable groups.

### 2.2. Iraq Case Study: 3iS Outreach to Improve COVID-19 Vaccination and Routine Immunization

#### 2.2.1. Background

COVID-19 vaccination uptake in Iraq is one of the lowest in the Middle Eastern region, for several reasons: vaccine hesitancy, disbelief in or perceived low threat posed by the virus in the country’s predominantly young population, weak disincentives for remaining unvaccinated, and mistrust in government [8]. COVID-19 vaccination coverage remains suboptimal in the country, with coverage of eligible populations (aged 12+ years) at ~38% for one dose and 27% for two doses as of December 2022 [9]. Less than 1% have received a third dose of COVID-19 vaccine [9]. Additionally, childhood routine immunization rates have been lagging due to hesitancy, fear of attending primary healthcare centers amidst a pandemic, exacerbation of traditional reluctance among some conservative communities, and poor access to services for other populations.

In addition, major efforts—beyond campaigns and supplementary immunization activities—are now required to catch-up on missed doses of routine immunizations, like measles and polio, due to the disruptions in health services from the COVID-19 pandemic. Consequently, it is important to integrate COVID-19 vaccination with the routine immunization program to increase the efficiency and impact of both programs.

#### 2.2.2. Approach

UNICEF partnered with the Ministry of Health of Iraq to develop the 3iS (Intensification of Integrated Immunization) outreach approach to enhance COVID-19 and routine immunization coverage and demonstrate a new approach to service delivery for basic primary care. The approach was launched in February 2022 and continues to date (see Figure 2 for a visual representation of the approach).

3iS aimed to accelerate control of COVID-19 by improving vaccine uptake, particularly amongst hard-to-reach and vaccine-hesitant groups and raise public awareness about COVID-19. The initiative also intended to bridge routine immunization coverage gaps and reach zero-dose children. Other objectives were to generate community demand for COVID-19 and other vaccination uptake and reduce the probability of vaccine-preventable disease resurgence and strengthen awareness of health sector services and health literacy among communities.

An outreach team was created in each primary healthcare center comprising one routine immunization vaccinator, one COVID-19 vaccinator, two registrars, one data entry officer, and one community mobilizer. The teams visited specific communities with a mobile clinic or set up vaccination points in village health houses, shrines, public parks, or affiliated camp clinics. The community mobilizer approached families at schools, shopping centers, and other key sites to discuss vaccination, answer queries, build trust, and encourage community members to visit the mobile clinic. Community mobilizers traveled house-to-house to identify and prime unvaccinated individuals before the vaccination team conducted its home visit. Community mobilizers and other team members were trained in interpersonal communication and drew on their familiarity with the communities to involve local leaders. Vaccinators were trained on the specifics of COVID-19 vaccination, including the handling of the different vaccines available.

The initiative also served to strengthen the capacity of the vaccination program more broadly, including efforts to strengthen related technical, communications, and information technology hardware capabilities, as well as develop the human resource capacity of health authorities at all levels across the country.

#### 2.2.3. Results

Through the 3iS campaign, over 4.4 million doses of COVID-19 and routine immunization vaccines were delivered between February and November 2022 across 155 districts in 19 provinces. The campaign is still ongoing in April 2023 and will continue quarterly. A key achievement was reaching around 149,000 zero-dose children who commenced their routine immunization schedule through the 3iS approach, dramatically reducing the risk of vaccine-preventable disease outbreaks in Iraq. Regarding COVID-19 vaccination, around 20% of all doses administered in 2022 were provided as part of 3iS activities across the country.

The enhancements to service delivery and intensive demand-creation efforts contributed to diphtheria, tetanus, and pertussis (DTP3) coverage for 2022 reported as 93% and measles at 88% by October 2022, which are the highest coverage rates since 1987 and 1999, respectively. In 2022, every province of Iraq had over 80% DTP3 coverage for the first time in at least eight years (Figure 3).

#### 2.2.4. Key Lessons Learned

The 3iS campaign strengthened routine immunization and emergency (COVID-19) vaccination planning, implementation monitoring and reporting in Iraq.

The campaign also deepened acknowledgement of health promotion as a key element of disease prevention and outbreak response, using direct community engagement by outreach to deliver health messages and encourage service uptake. The flexibility and locally rooted design of the 3iS campaign enabled community mobilizers to be versatile and engage communities on locally relevant topics, such as cholera prevention activities in areas where there was a risk of cholera resurgence.

The 3iS approach incurs costs of around US$1 million for outreach services per month of implementation, which is far less than the societal costs incurred for disease outbreaks. Similar approaches can be used in the future to promote universal health coverage of a wide range of preventative, promotive, and curative health services across the life cycle.

### 2.3. Ghana Case Study: Applying Human-Centered Design to Improve COVID-19 Vaccination Uptake

#### 2.3.1. Background

In early 2022, a large disparity existed between COVID-19 vaccine supply and coverage in Ghana. While the country had enough vaccines to inoculate 88% of its eligible population with at least one dose, uptake was notably low and inequitably distributed. Around half of the country’s available COVID-19 vaccines had been administered to about 16% of the target population. Findings from a pre-COVID-19 vaccination survey conducted in Ghana’s Greater Kumasi metropolitan area identified that around half of Ghanaians expressed an intention to get vaccinated against COVID-19 [10]. However, actual rates of COVID-19 vaccination have been low.

#### 2.3.2. Approach

Human-centered design (HCD) is a problem-solving process that begins with understanding the human factors and context surrounding a challenge and designing solutions that directly involve the users they aim to serve [11]. HCD is central to UNICEF’s work to find inclusive and person-centered solutions. To address vaccine-related issues in Ghana, UNICEF partnered with Common Thread Communications to develop a HCD initiative to improve COVID-19 vaccination uptake.

Over a three-day period, regional- and national-level stakeholders from Ghana Health Services, UNICEF staff, implementing partners, and civil society organizations received theoretical and practical HCD skills through activities facilitated by UNICEF and Common Thread Communications. These activities involved an initial introduction to behavioral design, a rapid enquiry in three locations, and an intervention co-design workshop informed by the rapid enquiry findings (Figure 4).

The rapid enquiry comprised a full day of observations and conversations with vaccinators and people receiving their COVID-19 vaccines in Kumasi. Three different vaccination sites were included in the rapid enquiry stage, including a workplace booster campaign at a Bank of Ghana Office, a static vaccination site at a government health center, and a mobile vaccination site at Kumasi City Market. Guiding questions used during the rapid enquiry included those such as “*How might we explain this gap between supply and vaccine coverage?*” and “*Which vaccination sites are most popular?*”.

#### 2.3.3. Results

From the rapid enquiry exercises, UNICEF, Common Thread Communications, and their partners quickly established that many people in this region were ‘vaccine opportunistic’. This meant that they did not have any particular vaccine hesitancy but were not urgently or actively seeking out vaccination. Most Ghanaians in the region were willing to get vaccinated if the opportunity arose (e.g., if vaccination was available at work, or at the market). This highlighted that for vaccine opportunistic people, it is critical to make the opportunity to get vaccinated salient and easy.

Findings from the rapid enquiry informed prototyping and iterating of wayfinding materials to direct people to convenient vaccination sites. Prototyping and feedback allowed for ideas to be tailored and improved by testing them with the community and frontline health workers. Table 3 outlines the critical issues addressed with communities and frontline health workers through the process, and Figure 5 shows an example of the messaging developed as part of the initiative.

#### 2.3.4. Key Lessons Learned

The HCD approach was useful for understanding barriers to COVID-19 vaccine uptake at a local level and developing tailored, locally relevant solutions to improve immunization and disease prevention initiatives. Prior to the intervention, very few mobile or static vaccination sites had any signs, posters, murals, or other visual indications that vaccines were available. Most community members relied on radio announcements or word of mouth to learn about vaccine availability, and within vaccination sites, the public relied on instructions from health workers to guide them through the vaccination process. The HCD approach identified key factors that helped design wayfinding to direct people to convenient vaccination sites.

### 2.4. India Case Study: Facilitating Community-Led Appropriate COVID-19 Behavior and Vaccination Linkages for Marginalized Communities across India through Risk Communication & Community Engagement

#### 2.4.1. Background

During the second wave of COVID-19, India witnessed an acceleration in the number of COVID-19 cases, reduced supplies of essential treatments and vaccines, as well as increased deaths and myths clouding the vaccination drive and leading to vaccine hesitancy [12,13]. In public health emergencies like the COVID-19 pandemic, RCCE is a key participatory approach to meaningfully engage communities facing a particular health-related threat to ensure that they have accurate, timely information, advice, and solutions [14,15] delivered through trusted sources and channels. RCCE has been one of the key pillars of UNICEF’s COVID-19 response [16].

A new initiative was also developed in India, known as COLLECT (Community-Led Local Entitlements & Claims Tracker) as a research and advocacy platform to fulfill the need for local evidence by creating a flow of information between those at the margins of society and authorities. In November 2021, the COLLECT-RCCE (C-RCCE) initiative began a six-month intervention focused on building community-level awareness on COVID-19 appropriate behavior across India, addressing myths related to vaccines, and boosting vaccination rates amongst vulnerable groups.

#### 2.4.2. Approach

The C-RCCE was a community-led initiative, supported by UNICEF India and implemented by the Praxis Institute for Participatory Practices (Praxis, India), a development support organization aimed at democratizing development processes to make them more inclusive, relevant, and responsive for marginalized communities. Between November 2021 and April 2022, the initiative was implemented across 11 states in India (Andhra Pradesh, Bihar, Chhattisgarh, Gujarat, Madhya Pradesh, Odisha, Rajasthan, Tamil Nadu, Telangana, Uttar Pradesh, and West Bengal), and within 70 districts, predominantly inhabited by Leave No One Behind Communities (LNOB). These LNOB comprised Dalit (lower caste), Adivasi (indigenous), Denotified (repealed categorization of [a tribe] as criminal under the Criminal Tribes Act enforced by the British Raj between l87l and I949 [17]) and Nomadic Tribes, and minority communities who have been further marginalized by the pandemic.

The overall program focuses on building a resource base at a community-level for easy access to information around vaccine hesitancy, myths, and access to primary healthcare and social protection. A system was also established to share information between households in the community and the local administration for prompt response and action, inclusive of community mobilization and interpersonal communication. This is particularly important for marginalized groups in the targeted areas, as their access to digital tools is often minimal, limiting their access to government public health services and schemes. Figure 6 shows examples of the types of posters and information shared as part of the project.

The program deployed 560 community-level fellows to gather information and insights on vaccination status, access to schemes and healthcare, and disseminate information to communities. Thirty-nine district-level coordination agencies and 31 district-level fellows were deployed to work with eight fellows, each. UNICEF and Praxis provided technical support for the engagement and worked with the district-level agencies. Baseline and endline surveys were conducted to assess the impact of the intervention, with the baseline survey conducted over twenty days between December 2021 and January 2022 and the endline survey conducted in April 2022. Copies of the surveys are available in the Appendix A.

#### 2.4.3. Results

Data were collected from 48,086 respondents in the baseline survey and from 44,900 respondents in the endline survey. Efforts were made to sample the same set of respondents across the baseline and endline surveys; however, due to migration and other factors, the respondent groups were not identical between rounds. The distribution of social categories is shown in Table 4.

Around 100,000 households (and 500,000 persons) from the most marginalized communities were reached directly through the program. The six-month engagement resulted in a 25% increase in the fully vaccinated population from baseline to endline, shown in Figure 7. While the results at aggregate level are uniform for both men and women, considering the socially disadvantaged position of women, the findings indicate that the program was effective in increasing vaccination rates among women. Other vulnerable groups such as persons with disabilities (PWD), pregnant women, and the transgender/non-binary population have reported increases in vaccination of 30%, 43% and 30%, respectively.

#### 2.4.4. Key Lessons Learned

Overall, collaboration with the community was critical to the success of the intervention, as was the use of a flexible, multi-pronged messaging medium. In addition, several specific components of the initiative were found to work well. This included engaging local volunteers or fellows from communities where interventions were taking place and engaging community-based organizations that were already working with these communities. Also important was the training and capacity building of volunteers and district coordinators around COVID-19 and COVID-19 appropriate behaviors and providing support to frontline health workers for vaccination drives. In addition, marginalized communities were provided with access to social welfare schemes. Collaborations paved the way for the intervention, notably with the panchayat (i.e., community governance council) and its representatives, including sarpanch (i.e., focal point of contact between government and community), ward members, district-level stakeholders, and teachers-in-charge/principals of upper primary and middle schools. Engagement with 39 community-led organizations identified the most vulnerable hamlets and households, comprising people with marginalized identities.

Critical to the success of the initiative was the use of a multi-pronged messaging medium to disseminate information through local networks, combined with flexibility in how this was deployed at the local level. The deployed messaging strategies took the form of intensive interpersonal communication and community meetings, educational and communications-based trainings with doctors, engaging local panchayats and key influencers as trusted health messengers, and mapping and addressing myths through regular follow-ups in community meetings. Door-to-door interactions with community members were also deployed to tackle the myths and rumors. As many parents/child caregivers were hesitant to get their children vaccinated, special awareness meetings and door-to-door awareness visits from health workers were organized with parents to provide correct information and reassurance around vaccination. This encouraged some parents to immediately take their children for vaccination.

Vaccine hesitancy was found to be driven by people’s lack of confidence, prevailing myths, misleading information, risk calculation and inconvenience to reach the vaccination centers. Rumors, myths, and misinformation about vaccines were particularly prevalent among Denotified and Nomadic Tribe communities. Availability of transport emerged as a major barrier to vaccination in tribal locations. Across all states, long queues without socially distant/safe waiting areas and the absence of ramps in vaccination centers created accessibility issues serving as barriers to vaccination for PWDs.

The information provided by community leaders and community-based organizations was used to inform specific interventions. Subsequently, special village-level vaccination camps and doorstep vaccination services were offered for PWD who were unable to access the main vaccination centers. To encourage vaccination among pregnant women, another key group, the team individually met husbands, and other family members to explain the importance, efficacy, and safety of COVID-19 vaccination.

## 3. Discussion

This article shares four case studies of how different community-driven initiatives were used to improve acceptance and uptake of COVID-19 vaccination and/or routine immunization across diverse and marginalized communities in India, Zambia, Iraq, and Ghana. These approaches—C-RCCE, community rapid assessments, 3iS, and HCD—can be replicated for immunization demand-creation among diverse and marginalized communities within LMICs.

Although different approaches were deployed across the four country interventions, all were rooted in an SBC approach aimed at understanding the needs of the community and collaboration at the community-level to remove barriers impeding uptake, and inform and develop solutions. RCCE is recognized as an essential component of responding to public health emergencies and has been central to UNICEF’s COVID-19 response. RCCE has previously been used as part of several emergency responses, including to manage the “infodemic” surrounding COVID-19, and as part of Ebola preparedness and response in Burundi, Rwanda, South Sudan, Tanzania and Uganda [18,19]. UNICEF in collaboration with partners of the RCCE Collective Service have developed global open-access toolkits and resources to support RCCE efforts that will support countries in future pandemic preparedness [16,20].

UNICEF recognizes that the interconnected nature of social norms, behavior, community systems, and health systems requires multidimensional tools to both diagnose root causes and intervene with a system of solutions. UNICEF employs the HCD approach to activate previously overlooked connections and leverage points, including nudging and normalizing immunization behaviour [11]. In recent years, the approach has been increasingly applied to co-design solutions that go beyond immunization to address other public health issues like maternal and child health interventions by centering the person and community experiencing the barrier or problem [21,22,23]. UNICEF has supported countries to institutionalize the use of HCD including integrating it into training curriculums of national health institutes, paving way for some Governments to establish behavioral science centers anchored on the HCD approach. UNICEF shares publicly available resources around how to use HCD to address challenges related to community demand for basic health services such as immunization [24,25].

These interventions also demonstrate the importance of applying tailored approaches that rely on behavioral evidence and demographic segmentation to understand and address the specific concerns of different groups. In Zambia, the deployment of rapid community assessment provided invaluable, timely insights into evolving attitudes around vaccination that allowed the identification of population segments for which tailored interventions could be developed. The experience in India also provided insights into community needs and drivers of vaccine hesitancy. The intervention demonstrated how marginalized communities need accurate health information during health emergencies to address their concerns and the importance of health authorities working with locally trusted sources to reach these groups.

Together, the case studies also highlight the value of multi-pronged approaches to tackle the complex, often interconnected factors affecting vaccine uptake. As demonstrated in Iraq’s intervention, COVID-19 RCCE can have even greater impact if positioned as part of a broader package of vaccination services targeted at improving wider routine immunization and primary health care.

These case studies showcase a wide variety of perspectives from diverse groups across differing geographical regions. This reduced the chance of introducing bias across case studies, with no group being privileged over another. Collaborative, multidisciplinary research, often with community members themselves, led to greater awareness of the wider social and political context of the setting.

Nevertheless, a limitation of the case studies is that they describe interventions conducted in distinct settings for specific populations. Consequently, approaches may need to be tailored when applied in other countries or even in different contexts within the same country, as populations and factors underlying acceptance of and demand for immunization services will vary between communities. However, the approaches described here, all rooted in SBC principles, provide different ways to identify community issues and develop community-driven solutions to improve vaccination rates and health service delivery more broadly.

Interventions were also sometimes affected by on-the-ground realities and challenges. For example, in Iraq, some communities were difficult to reach or inaccessible, particularly in areas formerly claimed by ISIS, and teams did not have access to some disputed areas. Instead, local health workers from tribal societies, who have experience working on the national polio and measles campaigns, lead campaigns in some of the affected areas. The approach for including these locations in the campaign was managed on a case-by-case basis by the Department of Health. In Zambia, specific limitations of the community rapid assessment approach included that there was a limited number of respondents from some provinces, which affected geographic comparison. To address this, multiple filters were applied to lift responses coming from such provinces, and prevent the allotted sample from quickly filling up with responses from big cities like Lusaka. In addition, as a mobile-based survey excludes those without mobile phones, findings were reviewed along with community feedback and insights from various community engagement platforms and interventions, and triangulated for action. In Ghana, the project was conducted in the Greater Kumasi area which imposes a limitation in terms of geographical coverage. Also, as an HCD approach is context-specific and needs to respond to the daily realities of end-users, there is a limitation in terms of generalization and replicating the specific approach in other settings. Thus, the outcome of the HCD approach described here can provide general lessons, but its adoption must be tailored to the specific context of new locations. Additionally, the HCD intervention will have contributed to but may not be solely responsible for the noted increases in COVID-19 vaccination.

## 4. Conclusions

This article shares lessons learned from four UNICEF interventions that supported Governments to generate acceptance and demand for COVID-19 and routine immunizations in Zambia, Iraq, Ghana, and India. Going forward, UNICEF’s work on SBC for immunization will continue to focus on promoting pro-equity, gender transformative, people-centered, and evidence-based tailored strategies to encourage demand and uptake of all vaccines, including COVID-19, as part of essential health services, while maintaining public trust in health systems and vaccines. Capitalizing on COVID-19 investments will be important to integrate COVID-19 vaccination demand into routine immunization and strengthen primary health care and community-based systems. Broader use of the tools and approaches shared here can be deployed to improve COVID-19 vaccination coverage, close gaps in routine immunization and strengthen pandemic preparedness efforts.

## Figures and Tables

**Figure 1 vaccines-11-01180-f001:**
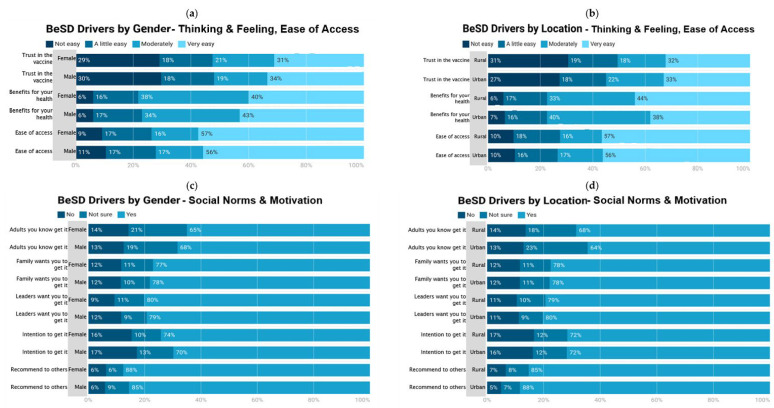
Drivers of vaccination. (**a**) by gender—thinking & feeling, ease of access; (**b**) by location—thinking & feeling, ease of access; (**c**) by gender—social norms and motivation; (**d**) by location—social norms and motivation.

**Figure 2 vaccines-11-01180-f002:**
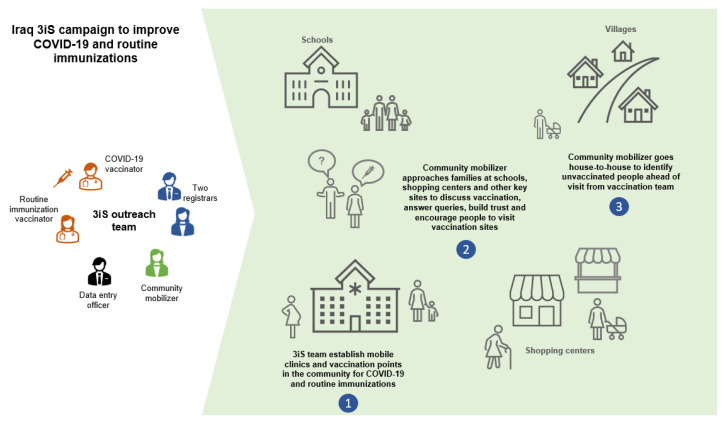
Iraq 3iS campaign approach.

**Figure 3 vaccines-11-01180-f003:**
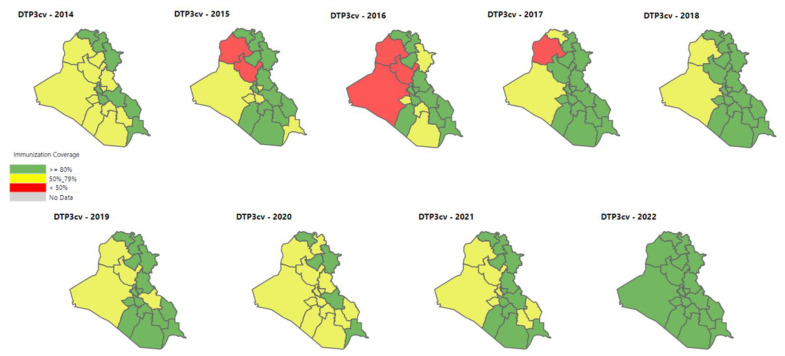
DTP3 coverage by province (2014–2022).

**Figure 4 vaccines-11-01180-f004:**
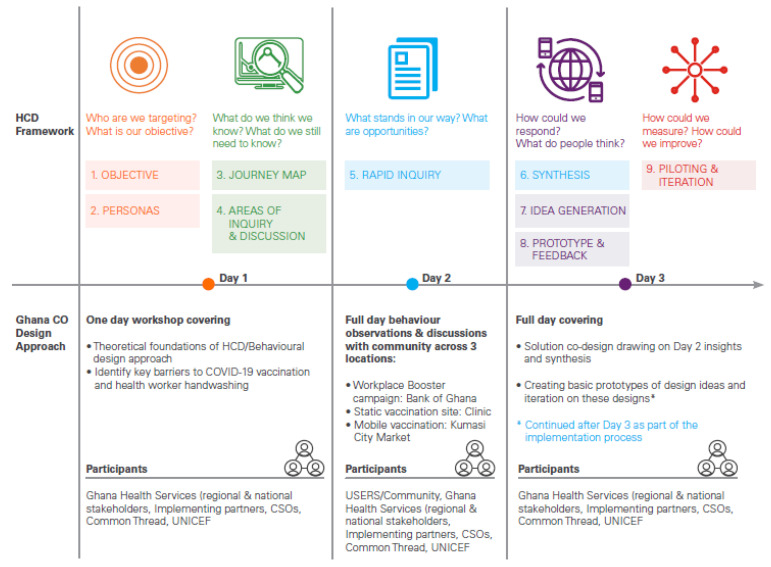
Human-centered design approach deployed in Ghana.

**Figure 5 vaccines-11-01180-f005:**
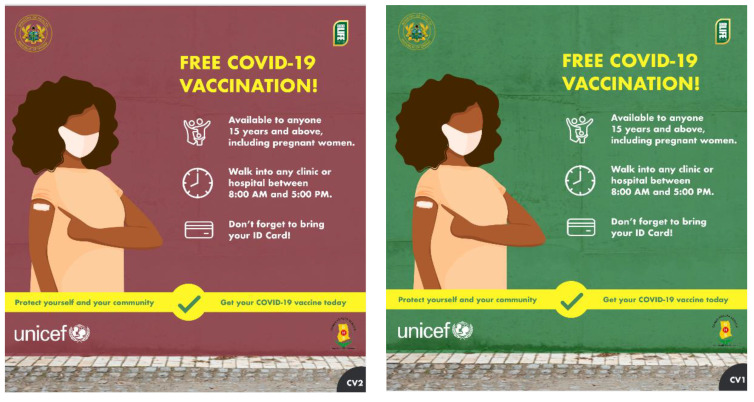
Example of messaging developed to highlight COVID-19 vaccination services in Ghana.

**Figure 6 vaccines-11-01180-f006:**
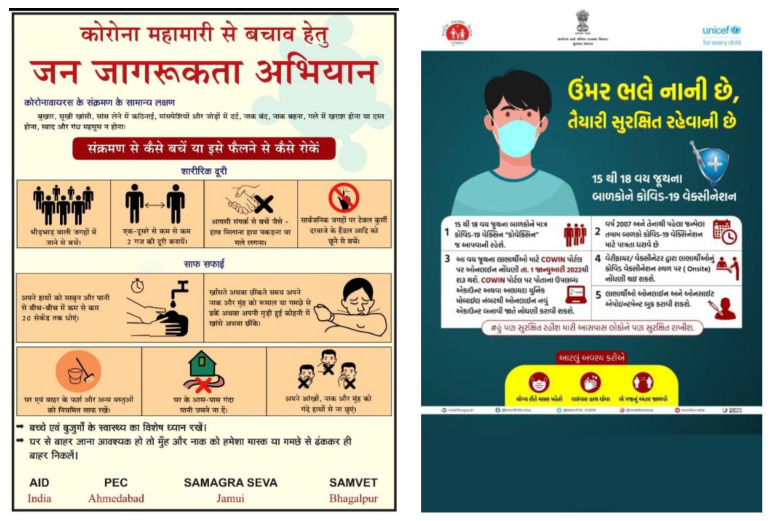
Examples of COVID-19 information posters in local languages shared as part of the project.

**Figure 7 vaccines-11-01180-f007:**
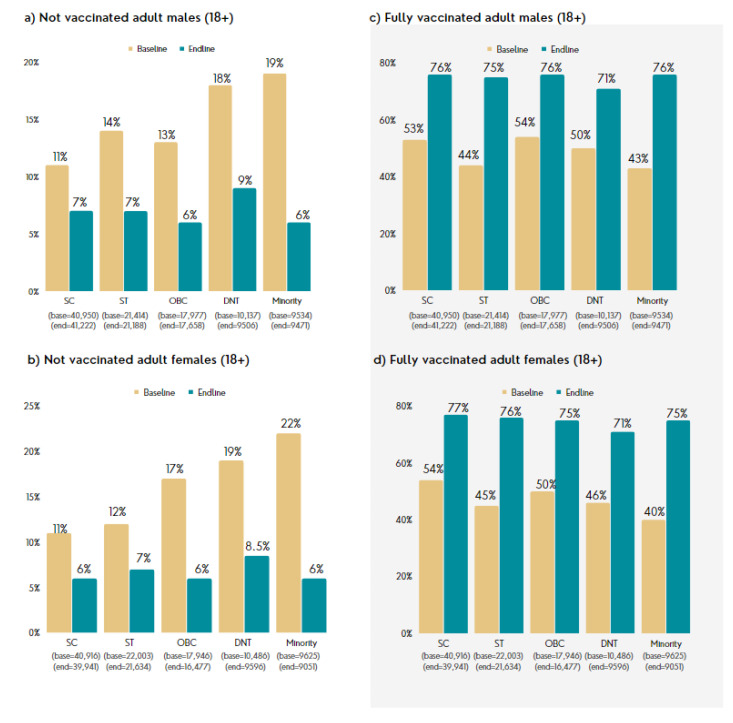
Change in vaccination rates between baseline and endline. (**a**) percentage of non-vaccinated adult males; (**b**) percentage of non-vaccinated adult females; (**c**) percentage of fully vaccinated adult males; (**d**) percentage of fully vaccinated adult females. DNT, Denotified and Nomadic Tribes; OBC, Other backward classes; SC, Scheduled caste; ST; Scheduled tribe.

**Table 1 vaccines-11-01180-t001:** Segments identified from community rapid assessment.

	Segment 1 Convert	Segment 2 Outsider	Segment 3 Dissenter	Segment 4 Analyst
Population share	R1 = 41.15% R2 = 57.33%	R1 = 19.95% R2 = 19.28%	R1 = 13.59% R2 = 5.19%	R1 = 25.32% R2 = 18.20%
Thinking & Feeling	Convinced of COVID-19 threat and not entertaining any doubts. Would quickly accept a COVID-19 vaccine if it would be easily accessible to them.	Entertaining neither doubts nor any knowledge. Often unsure about trust and any social processes, as if in a state of withdrawal, ready to endure whatever comes.	Not at all convinced by any broad intelligence on COVID-19 and vaccination against it. Highly susceptible to COVID-19 misinformation or doubts. Integrated denial.	Moderate in trust towards the COVID-19 vaccine. Likely to entertain multiple sources of information and divided on social norms. High in intent to accept vaccine.
Social Processes	Reasonably positive sense from others’ intentions.	Unsure and rather divided sense of others’ intentions.	Unsure or negative sense of others; little positivity.	Strong disagreement on others’ intentions.
Motivation	Nearly 100% on motivation and recommendation.	Very unsure or divided on self and other motivation.	Strongly discouraged on getting vaccinated.	Very positively motivated and recommending.
Practical Issues	Divided on the issue of ease of access.	Negative to moderate on ease of access.	Mostly negative on ease of access.	Negative to moderate on ease of access.

**Table 2 vaccines-11-01180-t002:** Specific social and behavior change interventions recommended for the different segments.

	Segment 1 Convert	Segment 2 Outsider	Segment 3 Dissenter	Segment 4 Analyst
Direction for social and behavior change interventions	Development of microplans to understand access-related barriers. Simplify registration processes. Increased outreach services in places and times that are convenient. for communities Lessen the financial burden of transport (e.g., increasing vaccination points and enhancing outreach in congregate settings such as religious institutions). Support elderly in accessing vaccination sites and their registration by partnering with youth organizations.	Ensure ease of access; clarify the what, where, and when. Showcase key influencers taking and promoting the vaccine with their personal stories and experiences. Make COVID-19 vaccination the default option (“it is time to get your COVID-19 vaccine” versus “would you like to be vaccinated?”). Disseminate messages that frame vaccination as the accepted social norm.	Maximize use of trusted sources and interactive/dialogue-based media. Partnering with health professional associations to influence the health worker community and the public. Timely sharing of accurate and fact-based information with tailored messages to help address safety concerns and information gaps. Engage religious leaders and other key influencers to strengthen the existing social environment.	Emphasize the importance of the shift from thinking about vaccine to the actual decision of getting it. Timely sharing of detailed and fact-based information on the working and benefits of COVID-19 vaccines. Maximize use of trusted sources and interactive/dialogue-based media. Increase tailored messages and interventions to help address safety concerns and information gap among vulnerable groups.

**Table 3 vaccines-11-01180-t003:** Critical issues addressed with communities and frontline health workers using prototyping, feedback and iteration.

COVID-19 Vaccination (Community Strategies)
Place branded signs outside each static and mobile vaccination site indicating the type of vaccines available, the days/hours of availability, eligible populations and requirements, and the approximate time needed to get a vaccine. Signs use consistent branding and should have a single visual indicating that vaccines are available at this site. Create signs, murals and posters within busy places in the community, including markets, indicating where to go for COVID-19 vaccination. Materials use consistent branding and symbols and include information on the days/hours of availability, eligible populations and requirements, and the approximate time needed to get a vaccine.

**Table 4 vaccines-11-01180-t004:** Social class distribution of survey respondents.

	Scheduled Caste	Scheduled Tribe	Other Backward Classes	Minority	Denotified and Nomadic Tribes
Baseline (*n* = 48,086)	49%	26%	20%	10%	11%
Endline (*n* = 44,900)	51%	26%	19%	11%	12%

## Data Availability

The data presented in this study are available on request from the corresponding author. The full data are not publicly available for confidentiality of the participants. For the Ghana case study, the entire data for the HCD work in Ghana was led by Common Thread Communications (a UNICEF Global Long Term Agreement holder on behavior science in collaboration with Ghana Health Services. Therefore, primary data for this project are in the custody of the UNICEF country office, Common Thread and Ghana Health Services. Data collected during the project should not be realized for any profit/business purpose.

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
