# Peer review of "Community-Based Approaches to Increase COVID-19 Vaccine Uptake and Demand: Lessons Learned from Four UNICEF-Supported Interventions"

_vaccines, 2023, doi:10.3390/vaccines11071180_

Round 1
Reviewer 1 Report
The subject is very interesting. The authors report on a study about four case studies and how different community-driver initiatives were used to improve COVID-19 vaccination coverage in India, Zambia, Iraq and Ghana.
I believe this is an important contribution to the literature, good enough for publication.
Author Response
Dear Editor,
Thank you very much for the time taken to review our article and the thoughtful comments we received. We are very grateful for your support and feel addressing the comments received has strengthened our article. We have addressed all comments in the listing below and trust that they will be to your satisfaction.
We look forward to your response and are hopeful about publication.
Best wishes
Review 1
The subject is very interesting. The authors report on a study about four case studies and how different community-driver initiatives were used to improve COVID-19 vaccination coverage in India, Zambia, Iraq and Ghana.
I believe this is an important contribution to the literature, good enough for publication.
Response: Thank you, we appreciate your feedback.
Reviewer 2 Report
Summary – I have read this manuscript with interest. This manuscript presents four case studies of community-based interventions to increase the uptake of COVID-19 vaccines. The topic is highly relevant and timely. The manuscript is well-written. I have a few major comments and the rest are mostly minor suggestions for improvements.
Major comments
1. As a case study, like all qualitative research, a reflexivity statement/paragraph from the authors detailing their characteristics, attributes, involvement with projects/participants, beliefs and backgrounds which may have influenced their work is required here.
2. Informed consent, IRB statement, Conflict of interest statement, and Data availability statement need to be completed
3. Limitations of the findings and interpretation of the findings should be added to the discussion section.
Minor Suggestions
1. Case study Zambia – It would be of interest to whether the authors note any pattern in the distribution of the four persona types along gender or location, and if so, what reasons do they hypothesize for this.
2. Case study Iraq – Line 216-218 – Citation needed
3. Case study Iraq – A graphical representation of the 3iS approach would improve the readability of this section.
4. Case study Ghana – Line 297-298 – Incomplete sentence, consider revision
5. Case study India – Examples of materials/pamphlets/posters used to disseminate information would be a valuable addition.
Case study India – Survey tools could be added as supplements to the manuscript
English language is fine barring minor typographical errors
Author Response
Summary – I have read this manuscript with interest. This manuscript presents four case studies of community-based interventions to increase the uptake of COVID-19 vaccines. The topic is highly relevant and timely. The manuscript is well-written. I have a few major comments and the rest are mostly minor suggestions for improvements.
Response: Thank you, we appreciate your feedback. Please find our responses to the points raised below.
Major comments
- As a case study, like all qualitative research, a reflexivity statement/paragraph from the authors detailing their characteristics, attributes, involvement with projects/participants, beliefs and backgrounds which may have influenced their work is required here.
Response: Thank you, we have added a reflexivity statement to the discussion section (lines 542–546).
“These case studies showcase a wide variety of perspectives from diverse groups across differing geographical regions. This reduced the chance of introducing bias across case studies, with no group being privileged over another. Collaborative, multidisciplinary research, often with community members themselves, led to greater awareness of the wider social and political context of the setting.”
- 2. Informed consent, IRB statement, Conflict of interest statement, and Data availability statement need to be completed
Response: We have updated these sections accordingly.
- Limitations of the findings and interpretation of the findings should be added to the discussion section.
Response: Thank you, we have noted the limitations of the findings to the discussion section on lines 547–576.
“Nevertheless, a limitation of the case studies is that they describe interventions conducted in distinct settings for specific populations. Consequently, approaches may need to be tailored when applied in other countries or even in different context within the same country as populations as well as factors underlying acceptance of and demand for immunization services will vary between communities. However, the approaches described here, all rooted in SBC principles, provide different ways to identify community issues and develop community-driven solutions to improve vaccination rates and health service delivery more broadly.
Interventions were also sometimes affected by on-the-ground realities and challenges. For example, in Iraq, some communities were difficult to reach or inaccessible, particularly in areas formerly claimed by ISIS, and teams did not have access to some disputed areas. Instead, local health workers from tribal societies, who have experience working on the national polio and measles campaigns, lead campaigns in some of the affected areas. The approach for including these locations in the campaign was managed on a case-by-case basis by the Department of Health. In Zambia, specific limitations of the community rapid assessment approach included that there was a limited number of respondents from some provinces, which affected geographic comparison. To address this, multiple filters were applied to lift responses coming from such provinces, and prevent the allotted sample from quickly filling up with responses from big cities like Lusaka. In addition, as a mobile-based survey excludes those without mobile phones, findings were reviewed along with community feedback and insights from various community engagement platforms and interventions, and triangulated for action. In Ghana, the project was conducted in Greater Kumasi area which imposes a limitation in terms of geographical coverage. Also, as an HCD approach is context specific and needs to respond to the daily realities of end-users, there is a limitation in terms of generalization and replicating the specific approach in other settings. Thus, the outcome of the HCD approach described here can provide general lessons, but its adoption must be tailored to the specific context of new locations. Additionally, the HCD intervention will have contributed but may not be solely responsible for the noted increases in COVID-19 vaccination.”
Minor Suggestions
- Case study Zambia – It would be of interest to whether the authors note any pattern in the distribution of the four persona types along gender or location, and if so, what reasons do they hypothesize for this.
Response: Thank you, we have added the following information to the Zambia results section to clarify the findings by gender and location:
“There were only small differences in findings between female and male respondents, suggesting little impact of gender on drivers of vaccination. A similar trend was seen across rural and urban participants, with few differences between respondents in urban and rural locations (Figure 1b). Together, the findings suggest that decision-making cuts across demographics.”
- Case study Iraq – Line 216-218 – Citation needed
Response: Thank you, we have added a reference to support this point.
- Case study Iraq – A graphical representation of the 3iS approach would improve the readability of this section.
Response: Thank you, we have developed a figure (new figure 2) to illustrate the 3iS approach.
- Case study Ghana – Line 297-298 – Incomplete sentence, consider revision
Response: Thank you, this sentence has been corrected.
- Case study India – Examples of materials/pamphlets/posters used to disseminate information would be a valuable addition.
Response: We have added examples of two posters used to share information around COVID-19 as Figure 6; please note that these are in local languages.
- Case study India – Survey tools could be added as supplements to the manuscript
Response: We have provided the survey tools as supplementary materials (S1-S4) and cited these in the text.

Reviewer 3 Report
Please consider the following points before considering the manuscript published in Vaccine:
1. Explain why the authors choose Zambia, Iraq, Ghana and India for the studies? What are the uniqueness of such countries? How about the other countries such as China?
2. Please clarify the types of vaccines received by the people living in these 4 countries. This is because there are too many independent variables for comparing these 4 countries and it is difficult to draw conclusion for the studies.
3. For the results of questionnairs, please try to perform cluster analysis to see whatever any similarity of the responds especially in case 1.
4. What are the assumptions of all case studies? For example data representative, uncertainty.....etc
5. For these 4 countries, can you suggest ONE positive control and ONE negative control for comparing the existing results. Apparently, the 4 cases illustrated by the authors cannot reflect which one is the best and which one is the worst among all countries.
6. I cannot find the common points for comparison in these 4 case studies. Can the authors find 1 to 2 common areas for analysis?
7. Please suggest further recommendations based on the existing studies otherwise it maybe meaningless to carry out such studies by putting these 4 countries together in these manuscript.
Th quality of English is good and it allows reader to read the manuscript easily.
Author Response
Please consider the following points before considering the manuscript published in Vaccine:
- Explain why the authors choose Zambia, Iraq, Ghana and India for the studies? What are the uniqueness of such countries? How about the other countries such as China?
Response: The countries were chosen as best examples of interventions recently conducted by UNICEF and partners to increase demand for and access to COVID-19 and routine immunizations. The specific countries were selected to highlight a range of different approaches rooted in social and behavior change principles. The countries were also selected to share how approaches have been used in regional diverse settings across low- and middle-income countries in Africa, Asia and the Middle East, although the approaches can be adapted to broader settings.
- Please clarify the types of vaccines received by the people living in these 4 countries. This is because there are too many independent variables for comparing these 4 countries and it is difficult to draw conclusion for the studies.
Response: The four countries deployed multiple COVID-19 vaccines based on COVAX guidelines and availability both of the vaccine products themselves as well as other conditions such as cold chain capacities in different settings. While there were issues around vaccine preferences impacting people’s willingness to vaccinate when vaccines first became available, these case studies share approaches and learnings around increasing COVID-19 vaccine demand and uptake in general, and the approaches are not specific to a particular vaccine product.
- For the results of questionnaires, please try to perform cluster analysis to see whatever any similarity of the responds especially in case 1.
Response: Unfortunately, we are not able to compare results from the different case studies as they were undertaken using different approaches and in different populations. The aim of the study was to highlight the different approaches and how they can be used to understand drivers of vaccine demand and uptake and develop solutions that are tailored to the specific setting. As we touch upon in the introduction and discussion sections, it is likely that the specific factors underlying acceptance of and demand for immunization services will vary between communities. Consequently, any intervention will need to identify specific issues (e.g., gender disparities in vaccination) and develop tailored solutions accordingly. This article intends to share case studies of how various social and behavior change approaches have been used in diverse settings to improve demand for and uptake of COVID-19 and routine immunizations, rather than identify common issues between countries.
- What are the assumptions of all case studies? For example data representative, uncertainty.....etc
Response: The case studies were not conducted as formal studies, but rather describe experiences of real-world interventions to improve vaccination rates. As such, it is difficult to provide specific assumptions. However, we have added a paragraph that discusses the limitations of the case studies more broadly to the discussion section.
- For these 4 countries, can you suggest ONE positive control and ONE negative control for comparing the existing results. Apparently, the 4 cases illustrated by the authors cannot reflect which one is the best and which one is the worst among all countries.
Response: The aim of our article is to share different approaches that have been used to increase vaccine uptake and demand used in different low- and middle-income countries. As such, it is not possible to discuss the approaches in terms of “best” and “worst”, as the optimal approach will depend on the specific setting, target population, and their needs. Instead, the article aims to share different ways of applying social behavior change approaches that can be used, combined, and adapted in other settings depending on the specific needs of the target population to improve vaccination rates.
- I cannot find the common points for comparison in these 4 case studies. Can the authors find 1 to 2 common areas for analysis?
Response: The case studies all applied social and behavior change approaches that blend scientific knowledge with community insights, to bring about positive change in COVID-19 and routine immunization rates in low- and middle-income settings. Although different approaches were deployed across the four country interventions, all were rooted in a social and behavior change approach aimed at understanding the needs of the community and collaboration at the community-level to inform and develop solutions. We touch upon this point in the introduction and discussion section.
- Please suggest further recommendations based on the existing studies otherwise it maybe meaningless to carry out such studies by putting these 4 countries together in these manuscript.
Response: Lines 476-481 and 548-556 summarize the usefulness and recommendations based on the case studies and associated approaches (C-RCCE, Community Rapid Assessments, 3iS, and HCD).
